# Research on the semantic segmentation of Thangka images via an improved PIDNet

**Jiao Wu[1], Tiejun Wang [2,3]\*, Xiaoyan Hu[2,3], LingMei Tao[2,3], Tianjiao Duan[1], Yanjiao Wei[1]**

**1** Key Laboratory of Language and Cultural Computing of the Ministry of Education Northwest Minzu University, Lanzhou, Gansu, China, **2** School of Mathematics and Computer Science Northwest Minzu University, Lanzhou, Gansu, China, **3** Gansu Provincial Engineering Research Center for Multimodal Artificial Intelligence, Lanzhou, Gansu, China

\* wtj@mail.lzjtu.cn

## Abstract

In the digital preservation, restoration, and research of Thangka paintings, real-time semantic segmentation plays a crucial role in rapid image analysis. However, Thangka images exhibit intricate compositions, where principal figures often blend with backgrounds, ritual objects, and intricate ornaments, leading to blurred boundaries and fine details that challenge conventional segmentation methods in balancing accuracy and efficiency. To address this, we propose an improved PIDNet-based model incorporating ECA-Pag (Efficient Channel Attention-Path Aggregation) and LGFM (Local-Global Feature Fusion Module) modules, along with PConv3 (Partial Convolution 3), enhancing feature extraction and segmentation precision. Experimental results demonstrate that our model achieves 73.28% mIoU and mB-Fscore (mean Boundary F-score) of 40.01% on a custom Thangka dataset while maintaining 109.03 FPS (Frames Per Second), ensuring both high accuracy and real-time performance. Furthermore, evaluations on the Cityscapes benchmark confirm the model's generalization capability, outperforming baseline methods. This work provides an efficient and reliable solution for Thangka image segmentation, with potential applications in cultural heritage preservation and broader computer vision tasks.

## 1. Introduction

Semantic segmentation, a key task in computer vision, aims to assign each pixel in an image to a corresponding semantic category. Among the numerous application scenarios, the semantic segmentation of Thangka images presents unique practical significance and research challenges. Thangkas, traditional Tibetan painting artworks, contain rich details and complex patterns imbued with profound cultural information [1]. Semantic segmentation of Thangka images enables the accurate identification of elements, such as figures, costumes, and backgrounds, providing a foundation for digital preservation. The segmentation results can also serve as references for restoration, assisting restorers in more precisely reconstructing damaged

**Data availability statement:** We have uploaded the minimal anonymized dataset (including original Thangka images, segmentation masks, annotated JSON files, and pre-trained model weights) necessary to replicate the study findings to Zenodo, a stable public repository. The details are as follows: •Repository Name: Zenodo •Dataset DOI: 10.5281/zenodo.19498530 •Dataset Link: https://zenodo.org/doi/10.5281/zenodo.19498530.

**Funding:** This work was supported by the National Natural Science Foundation of China (Grant No. 62166035, recipient: Mrs. Tiejun Wang), the Fundamental Research Funds for the Central Universities (Grant No. 31920240091, recipient: Mrs. Tiejun Wang), and the Natural Science Foundation of Gansu Province (Grant No. 25JRRA991, recipient: Mrs. Tiejun Wang).

**Competing interests:** The authors have declared that no competing interests exist.

parts. This not only contributes to the protection and inheritance of this precious cultural heritage but also provides crucial technical support for Thangka research, restoration, and digitization.

With the rapid development of computer vision technology, semantic segmentation, as its core task, has been widely applied in autonomous driving, medical image analysis, cultural heritage protection, and other fields [2]. Real-time semantic segmentation technology, in particular, has become a research hotspot because of its ability to maintain high computational efficiency while ensuring segmentation accuracy. In the field of cultural heritage protection, real-time semantic segmentation significantly improves image processing efficiency, providing efficient and precise image information support for subsequent restoration and protection work.This real-time capability is essential for interactive digital archiving systems or on-site analysis during restoration projects, where immediate feedback is required.

In recent years, numerous researchers have proposed various efficient real-time semantic segmentation models, such as DDRNet [3], STDC [4], and PIDNet [5]. These models significantly increase the inference speed while maintaining high accuracy, meeting the real-time requirements of practical applications. Wang T et al. conducted preliminary semantic segmentation experiments on Thangka images via a Thangka semantic segmentation dataset, achieving good segmentation results, but the model's computational efficiency remains low [6]. Therefore, in-depth research on efficient and real-time semantic segmentation methods holds both essential theoretical value and practical significance for promoting applications in related fields.

To further improve the accuracy and real-time performance of semantic segmentation, researchers have proposed various improvement methods. In terms of model architecture, introducing attention mechanisms to enhance the model's perception of key regions is an important research direction [7–10]. For example, the model proposed by Liu Y et al. [11], which combines channel and spatial attention mechanisms, effectively enhances the model's perception of key regions. In terms of feature extraction, the multi-scale detail fusion networks proposed by Zhang Z et al. [12] and Li T et al. [13] can preserve fine image features while improving segmentation performance. Additionally, real-time semantic segmentation methods that combine multi-branch structures and contextual information [14] perform well in complex scenes. However, in Thangka images, owing to their rich background information and complex textures, models still face significant difficulties in identifying key elements, such as prominent figures and ritual instruments, as shown in Fig 1.

In recent years, with a deeper understanding of the relationship between local and global contexts in semantic segmentation tasks, researchers have gradually combined contextual information with multi-scale features to increase model performance [15,16]. Contextual contrastive learning for semantic segmentation [17,18] enhances the model's understanding of the relationship between local and global features, thereby improving segmentation accuracy. Moreover, UNetFormer [19] significantly enhances the recognition of object boundaries in complex scenes by introducing multi-scale features and boundary-aware mechanisms. Furthermore, studies on multi-scale context and boundary awareness in lightweight camouflaged

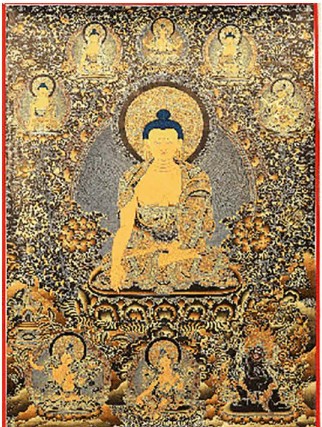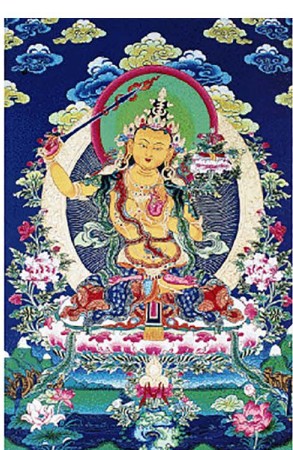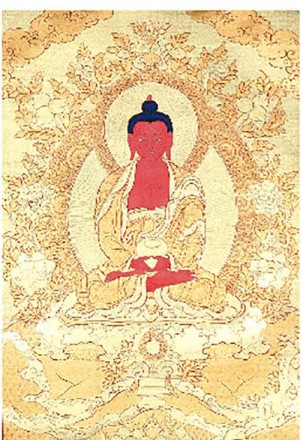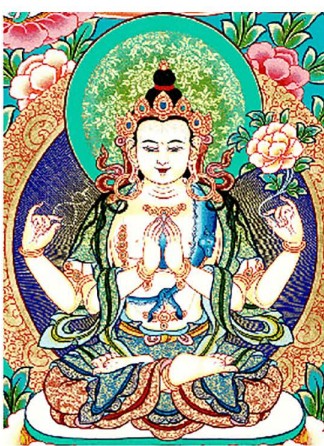

**Fig 1. Thangka image.** [a]The image shows typical Thangka composition characteristics, including a central deity, ritual instruments (e.g., vajra, lotus flower), and intricate background patterns.

object detection [20] also indicate that combining multi-scale information and boundary awareness is crucial for enhancing segmentation performance in complex texture backgrounds. Models fusing multi-scale contextual features and attention mechanisms based on convolutional neural networks have achieved good results in semantic segmentation tasks [21–23], further verifying the effectiveness of multi-scale and attention mechanisms in specific tasks.

The existing methods still have certain limitations when processing such detailed images. For example, attention-based methods, although they perform well in natural photos, struggle to balance local details and global semantics in Thangka images. As a typical multi-branch semantic segmentation network, PIDNet extracts features at different scales through three branches: P, I, and D. Branch P focuses on local details, branch I fuses multi-scale semantic information through the PPM(Pyramid Pooling Module) module, and branch D is responsible for global semantics. To capture the characteristics of Thangka images—complex backgrounds, diverse textures, and rich details—this paper proposes a real-time semantic segmentation model enhanced on the basis of the PIDNet network structure. The specific research contents and innovations are as follows:

1. Construction of a high-quality dataset for Thangka image semantic segmentation: The dataset includes 11 ritual instrument semantic categories and one main figure category. It contains 783 high-quality Thangka images, all of which are manually annotated to ensure accuracy and consistency. To further enhance the model's generalization ability, the dataset is expanded to 4,698 images through Thangka-specific augmentation. The construction of this dataset provides necessary data support for Thangka image semantic segmentation research and lays a solid foundation for subsequent model training and evaluation.

2. Proposed LGFM: This module extracts and fuses features through local and global branches, effectively capturing detailed information and contextual information in Thangka images to improve semantic segmentation accuracy. The local branch performs feature extraction and fusion through a key map, query, and value map, similar to the Transformer architecture [24], whereas the global branch generates an attention map via Dconv (depthwise separable convolution) [25] and attention mechanisms. The feature maps from the two branches are fused to create the final output features.

3. Improvement of the P branch in the PIDNet network: The original Pag module, which fuses information from the I and P branches, is improved to the ECA-Pag module, further enhancing the model's perception of key regions. Moreover,

LGFM is added to the end of the P branch to enhance the model's feature extraction and fusion capabilities, improving semantic segmentation accuracy.

4. Modification of the D branch structure: The D branch (detail branch) is mainly responsible for capturing fine details in images, which is particularly important for processing Thangka images with complex textures and details. Therefore, Partial Convolution (PConv3 [26]) is introduced before the add step in the D branch, further enhancing the model's ability to capture details. Partial convolution also improves computational efficiency, maintaining the model's lightweight nature.

Through the above improvements, the model proposed in this paper achieves significant performance improvements in semantic segmentation tasks on the self-built Thangka image dataset, providing strong technical support for Thangka image research and protection.

## 2. Methods

### 2.1. Overview

PIDNet (Parallel, Interleaved, and Dilated Network) is a deep learning model tailored for real-time image segmentation, which realizes efficient and accurate feature extraction and segmentation via parallel, interleaved and dilated convolutions. Its core idea is to capture multi-scale and multi-level features through a multi-path structure, thus balancing segmentation accuracy and real-time inference performance. The model consists of three core paths: the Proportional (P) path, Integral (I) path and Derivative (D) path. The P path parses detailed information from high-resolution feature maps to capture fine image details; the I path aggregates local and global contextual information; the D path extracts high-frequency features (especially object boundaries) to accurately segment different targets and further boost model performance.

The overall structure of the improved PIDNet model is shown in Fig 2. First, the input image is processed by multiple convolutional layers for initial feature extraction. For the P branch, the ECA-Pag module is first used to enhance the accuracy of feature representation, followed by the LGFM module for deep fusion of local and global features. For the D branch, PConv partial convolution is introduced to extract fine image details, which automatically ignores invalid or occluded regions and focuses on valid information, thereby strengthening the model's capability of capturing fine-grained

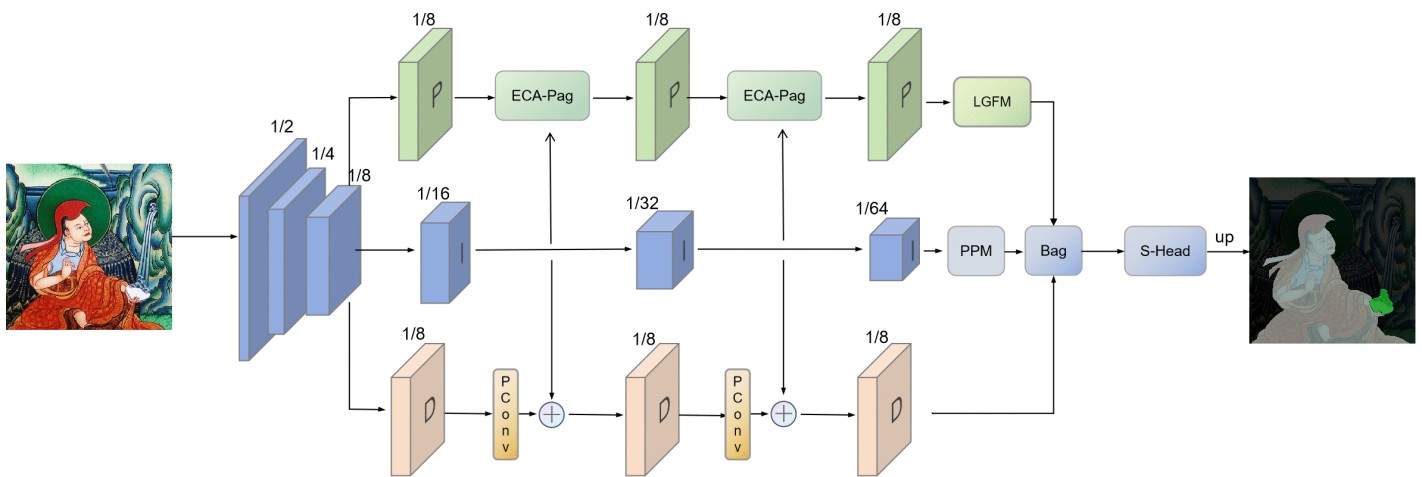

**Fig 2. Overall framework diagram of the model.** [a]The model integrates ECA-Pag, LGFM, and PConv3 modules based on the original PIDNet. The P branch incorporates ECA-Pag and LGFM for enhanced feature fusion, while the D branch adds PConv3 for fine detail capture. Input image size: 512×512 pixels; output: semantic segmentation map with 12 categories (1 main deity + 11 ritual instruments).

features. Finally, the Softmax classification head classifies the fused features to output the final segmentation map. Through the above optimizations, the improved PIDNet model achieves excellent performance in Thangka image semantic segmentation, with outstanding advantages in fine detail extraction, global contextual information capture and feature fusion, providing a reliable technical solution for Thangka image semantic segmentation.

## 2.2. ECA-pag module

The P branch of PIDNet captures local detail information, and to enhance its feature extraction and fusion capabilities, we introduce the ECA-Pag and LGFM modules into it.

The ECA-Pag module is illustrated in Fig 3. On the basis of the original Pag module, the ECA-Pag module is formed by incorporating the ECA module [27] (Efficient Channel Attention Module) and introducing residual connections. The ECA module enhances the model's perception of key regions by adaptively adjusting the importance of different channels. Specifically, the ECA-Pag module can dynamically change the weights of each channel according to the statistical information of the feature map, highlighting important feature channels. For Thangka images, this enables the model to more accurately capture and understand their rich and unique local details, such as the edges of prominent figures in Thangkas and the exquisite patterns of ritual instruments, improving semantic segmentation accuracy.

At the end of the P branch, the LGFM module is added. The LGFM module further enhances the model's feature extraction and fusion capabilities by combining local and global attention mechanisms, enabling the model to better handle the complex compositions and diverse elements in Thangka images and comprehensively grasp image information.

## 2.3. LGFM module

In semantic segmentation tasks, effectively fusing local detail information and global contextual information is crucial for improving model performance. Shuai Hu et al. [28] proposed the CAFM module, a fusion module that combines convolution and attention mechanisms. It extracts local features through convolution operations and global features through attention mechanisms and then fuses these two types of features. However, when extracting local features, the CAFM module relies only on convolution operations. It fails to fully utilize the rich contextual information between adjacent keys, which limits its self-attention learning ability on 2D feature maps to a certain extent.

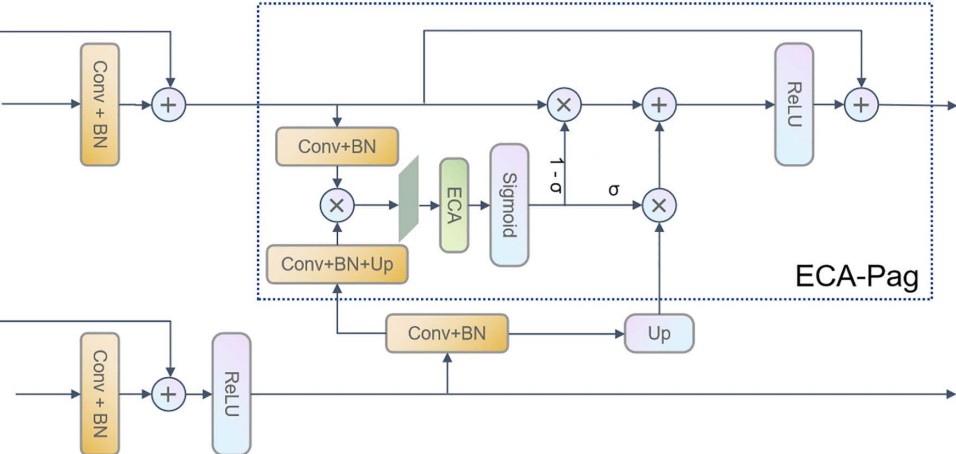

**Fig 3. The ECA-Pag Module Diagram.** [a]The module integrates the ECA (Efficient Channel Attention) mechanism and residual connections into the original Pag (Path Aggregation) module. It adaptively adjusts channel weights to enhance the model's perception of key regions (e.g., Thangka deities and ritual instruments).

To address this issue, the LGFM module proposed in this study improves upon the CAFM by introducing the COT [29] module to replace the local attention mechanism in the CAFM. The COT module not only retains the advantages of the self-attention mechanism but also fully utilizes the rich contextual information between adjacent keys, thereby achieving more effective self-attention learning on 2D feature maps. Through this improvement, the LGFM module can not only extract more accurate local features but also achieve a deeper fusion of local and global features by introducing global features, thereby enhancing the model's feature representation ability. The specific model structure is shown in Fig 4.

Specifically, let the input feature map be $X \in R^{C \times H \times W}$, where $C$ is the number of channels, $H$ is the height, and $W$ is the width. The LGFM module processes the input $X$ through two branches: the local branch and the global branch.

In the local branch, the query vector $Q$, key vector $K$, and value vector $V$ are first generated through linear transformations, as shown in equation (1):

$$Q = W_Q^T X, \quad K = W_K^T X, \quad V = W_V^T X \tag{1}$$

where $W_Q, W_K, W_V \in R^{C \times C'}$ are the corresponding weight matrices.

Next, the key vector $K$ undergoes a $k \times k$ convolution operation, and then the query vector $Q$ is concatenated with the convolved key vector $K$ to obtain $Z$, as shown in equation (2):

$$Z = \text{Concat}(Q, \text{Conv}_{k \times k}(K)) \tag{2}$$

A $1 \times 1$ convolution kernel $\theta$ first transforms the concatenated result $Z$ to obtain $A$, and then $A$ is processed by a $1 \times 1$ convolution kernel $\delta$ to obtain $B$, as shown in equation (3):

$$A = \theta(Z) = W_\theta Z, \qquad B = \delta(A) = W_\delta A \tag{3}$$

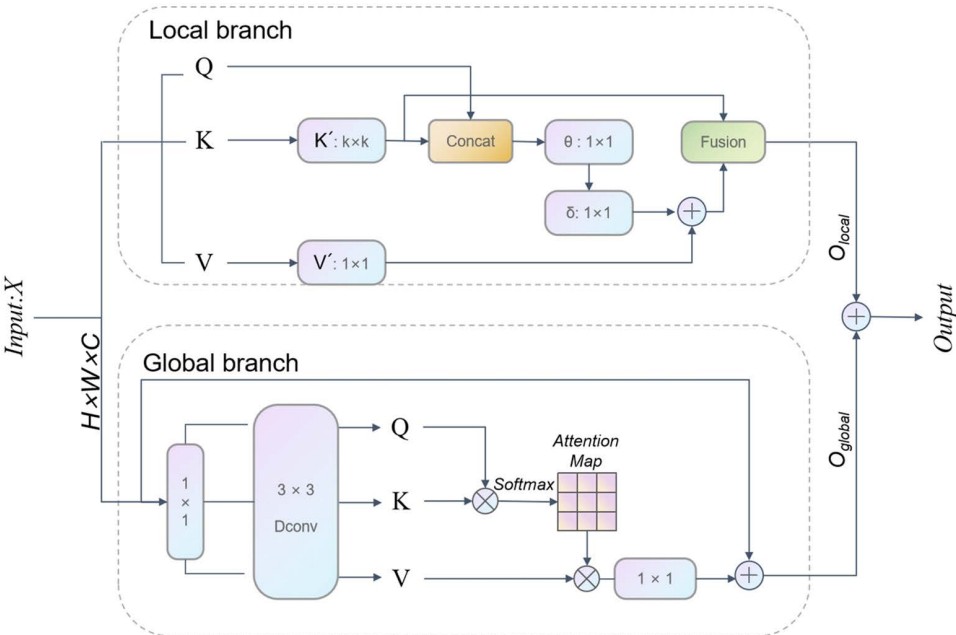

**Fig 4. LGFM Module Diagram.** [a]The LGFM module consists of local and global branches. The local branch uses COT-based self-attention to capture fine-grained details, while the global branch uses dilated convolution and attention mechanisms to model contextual information. Input: feature map X (C×H×W); output: fused feature map O (C×H×W).

where $W_\theta, W_\delta \in R^{C \times C'}$ are the weight matrices of the $1 \times 1$ convolutions.

The fusion operation is performed on B and V after a $1 \times 1$ convolution to obtain the output $O_{local}$ of the local branch, as shown in equation (4):

$$O_{local} = B + Conv_{1\times1}(V) \tag{4}$$

For the global branch, the input feature map X first undergoes a $1 \times 1$ convolution, followed by a $3 \times 3$ dilated convolution (Dconv) to generate the query vector Q, key vector K, and value vector V, as shown in equation (5):

$$Q, K, V = Dconv_{3\times3}(Conv_{1\times1}(X)) \tag{5}$$

where $Dconv_{3\times3}$ represents a $3 \times 3$ dilated convolution.

Generation of the attention map: The query vector Q and key vector K are dot-produced, and then the attention map S is generated through the Softmax function, as shown in equation (6):

$$S = Softmax(\frac{Q^T K}{d_k}) \tag{6}$$

where $d_k$ is the dimension of the key vector.

The attention map S is multiplied by the value vector V, followed by a $1 \times 1$ convolution and adding the original features to obtain the output $O_{global}$ of the global branch, as shown in equation (7):

$$O_{global} = Conv_{1\times1}(S * V) + X \tag{7}$$

Finally, the output $O_{local}$ of the local branch and the output $O_{global}$ of the global branch are added to obtain the final output O of the LGFM module, as shown in equation (8):

$$O = O_{local} + O_{global} \tag{8}$$

The introduction of LGFM realizes the deep fusion of local and global features, enhancing the model's feature representation ability. It is integrated into the end of the P branch of the PIDNet semantic segmentation network to further improve the segmentation accuracy. To interpret the module's effectiveness, we visualized the attention maps of the LGFM module (see Supplementary S1 Fig). The results show that LGFM effectively focuses on key regions (e.g., ritual instruments and deity edges) while suppressing background interference, verifying its role in feature enhancement.

## 2.4. PConv3

The D branch in PIDNet is mainly responsible for capturing fine details in images. PConv3 is a partial convolution operation that can retain more effective information when processing complex textures and details in images. Therefore, the PConv3 partial convolution operation is introduced, and its structure is shown in Fig 5.

Specifically, PConv3 extracts local features from images through partial convolution operations and effectively integrates these features through feature fusion operations. The design of this module is based on the concept of partial convolution, which aims to consider only a portion of the input features during convolution to reduce computational complexity and enhance efficiency.

The input to the module is a feature map X with dimensions $H \times W \times C$, where H and W represent the height and width of the feature map, respectively, and C represents the number of channels. To implement partial convolution, a mask M is

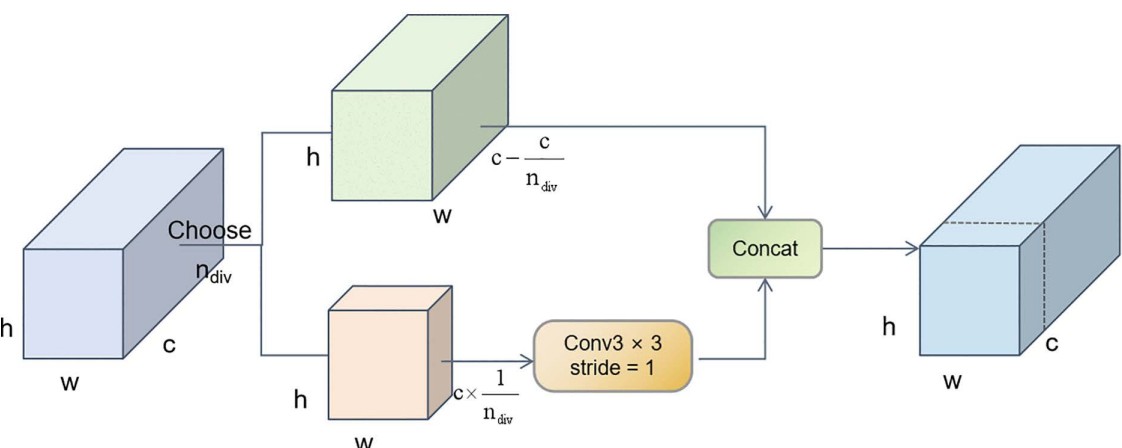

**Fig 5. The PConv3 Module Diagram.** ªPConv3 uses a mask-based partial convolution strategy to extract local features. It ignores invalid/occluded regions and focuses on effective detail information, which is critical for capturing intricate patterns in Thangka images. Input: feature map X (H×W×C); output: refined feature map Z (H×W×C).

first generated, with the exact dimensions as the input feature map X..The mask M is used to indicate which positions of the input features should be considered. The mask M can be generated via the following formula, as expressed in equation (9):

$$M = \sigma(W_m * X + b_m)$$

(9)

where $\sigma$ is the Sigmoid activation function and where $W_m$ and $b_m$ are the weights and biases of the mask generation network, respectively.

After generating the mask M, partial convolution operations are performed. Partial convolution weights the input feature map X through the mask M, as shown in equation (10):

$$Y = (W * (X \odot M)) \oslash (W * M + \epsilon)$$

(10)

where W is the weight of the convolution kernel, $\odot$ represents element-wise multiplication, $\oslash$ represents element-wise division, and $\epsilon$ is a small constant to avoid division by zero.

The output Y of the partial convolution undergoes a convolution operation to generate the final output feature map Z, as shown in Equation (11):

p

$$Z = W_z * Y + b_z$$

(11)

where $W_z$ and $b_z$ are the weights and biases of the output convolution layer.

PConv3 enables PIDNet to capture fine details in Thangka images, thus improving the accuracy and robustness of semantic segmentation.

## 2.5 Experimental Setup

The experimental environment utilized a high-performance computing server equipped with an Intel Xeon Gold 6240 processor, 256 GB of memory, and an NVIDIA RTX 3090 graphics card. For software implementation, the mmsegmentation

platform [30] was utilized in conjunction with the Python programming language and the PyTorch deep-learning framework.

The model was optimized using the Adam optimizer with an initial learning rate of 0.01, weight decay (L2 regularization) of $1 \times 10-4$, and a batch size of 8. To prevent overfitting, we employed L2 weight decay on all trainable parameters and inserted a dropout layer (rate = 0.1) before the final classification head. For early stopping, 755 images (approximately 20%) were reserved from the training set as a validation set. Under these configurations, the average training time for the proposed model was approximately 1 hour, with a peak GPU memory consumption of roughly 6.5 GB, ensuring efficient deployment capabilities.

## 2.6 Dataset

Our custom Thangka semantic segmentation dataset comprises 783 high-resolution images, each manually annotated at the pixel level across 12 categories: one main deity and 11 ritual instruments. The final dataset includes annotated categories of ritual implements comprising vajra, treasure vase, alms bowl, dharma wheel, lotus flower, scripture, vajra bell, kapala bowl, sword, umbrella, and khakkhara. Each annotation file is saved in JSON format using LabelMe, recording the category label of every pixel. The specific quantity corresponding to each type of ritual implement is presented in Table 1.

To preserve critical compositional elements—such as the deity's head and upper-body ritual objects—all original images were vertically shifted upward by 15% prior to cropping to a uniform size of $512 \times 512$ pixels. Considering the diversity of Thangka artistic styles and potential degradations due to storage conditions and digitization (e.g., fading, blur, sensor noise), we applied photometric data augmentation during training to improve model robustness. The augmentation strategies include brightness adjustment, contrast enhancement, saturation variation, additive Gaussian noise injection, and image sharpening. These transformations simulate realistic variations in real-world Thangka images without altering geometric structure or semantic layout. After augmentation, the effective training set size increased to 4,698 samples. The full dataset was split into training and testing sets at an 8:2 ratio, yielding 3,776 training images and 922 test images. The test set was held out completely and never used for training, validation, or hyperparameter selection.

## 2.7 Evaluation Metrics

In semantic segmentation tasks, the mIoU [31] serves as a key metric for evaluating model segmentation performance and quantifying the overlap between the predicted and actual segmentation regions. The intersection over union (IoU) is calculated as shown in Equation (12):

$$IoU_i = \frac{TP_i}{TP_i + FP_i + FN_i}$$

(12)

where TPi represents the number of actual positive pixels for the i-th category (pixels correctly identified as that category), FPi represents the number of false positive pixels (pixels incorrectly identified as that category), and FNi represents the number of false negative pixels (pixels that should have been identified as that category but were not). The mIoU is obtained by calculating the average of the IoU values for all categories, as shown in Equation (13):

$$mIoU = \frac{1}{N}\sum_{i=1}^{N} IoU_i$$

(13)

**Table 1. Categories of Dharma Instruments and Their Corresponding Quantities in the Dataset.**

| Dharma Instrument Name | Vajra | Vase | Bowl | Dharma Wheel | Lotus | Scripture | Vajra Bell | Kapala Bowl | Sword | Umbrella | Khatvanga |
|---|---|---|---|---|---|---|---|---|---|---|---|
| Quantity | 81 | 65 | 260 | 36 | 392 | 115 | 59 | 27 | 113 | 12 | 14 |

Here, N denotes the total number of categories. In the semantic segmentation of Thangka images, accurately and precisely segmenting central figures and ritual instruments is crucial. Given the relatively stable positions of these objects, the mIoU effectively reflects the model's segmentation performance for these targets. The model demonstrates exemplary performance in segmentation tasks by understanding the spatial relationships between central figures and ritual instruments and extracting relevant features, particularly in handling the complex edges and shapes of ritual instruments.

Additionally, this study introduces the mB-Fscore [32](mean Boundary F-score)as a new evaluation metric to measure the model's boundary segmentation capability for objects comprehensively. The mB-Fscore (mean Boundary F-score) is the average of boundary F-scores for all categories, effectively reflecting the balance between precision and recall in boundary localization. The calculation process is as follows: First, the boundary regions of the prediction results and the ground truth are extracted through dilation operations. Then, the boundary IoU between the predicted and true boundaries is calculated. Finally, the boundary F-score for each category is computed on the basis of precision and recall. The final mB-Fscore (mean Boundary F-score)is the average of the boundary F-scores for all the categories. The relevant calculations are shown in Equation (14):

p

$$mB - Fscore = \frac{1}{N} \sum_{i=1}^{N} B - Fscore_i \tag{14}$$

where N is the total number of categories and where $B - Fscore_i$ represents the boundary F score for the i-th category, which is calculated as follows:

$$B - Fscore_i = \frac{2 \times \text{Boundary IoU}_i}{\text{Boundary IoU}_i + 1} \tag{15}$$

where $\text{BoundaryIoU}_i$ is the boundary intersection over union for the i-th category, computed as in equation (16):

$$\text{BoundaryIoU}_i = \frac{\text{intersect}_i}{\text{union}_i + \varepsilon} \tag{16}$$

In this equation, $\text{intersect}_i$ is the number of intersecting pixels between the predicted and actual boundaries for the i-th category, $\text{union}_i$ is the number of union pixels, and $\varepsilon$ is a small constant (typically set to 10--6) to prevent division by zero.

Model parameter count and FPS are also used to evaluate the model's efficiency in terms of lightness and computational requirements.

## 2.8 Experimental design

To comprehensively evaluate the effectiveness of the proposed Thangka image semantic segmentation method, two sets of key experiments were designed: comparative experiments and ablation experiments. Comparative experiments were conducted to verify the superiority of the proposed method over state-of-the-art real-time semantic segmentation models on the custom Thangka dataset; ablation experiments were used to validate the contribution of each key module to the final model performance.

All models adopted a unified training configuration for fair comparison: Adam optimizer, initial learning rate of 0.01, batch size of 8, and a maximum of 45,000 training iterations. To ensure statistical rigor and reproducibility, each experimental configuration was independently repeated 5 times with different random seeds, and the final quantitative results are reported as the mean±standard deviation. Core evaluation metrics including IoU, mB-Fscore, number of model parameters and FPS were used to comprehensively measure the segmentation accuracy and inference efficiency of the models.

The Thangka image dataset and the source code of the proposed model will be made fully publicly available at https://github.com/JikD-591/thangka-seg upon the acceptance of this paper. The final repository will include a detailed dataset description, complete training/validation/test scripts, pre-trained model weights, and experimental configuration files to ensure the reproducibility of all experimental results reported in this paper. The collection and analysis of the Thangka image dataset comply with the terms and conditions of the original data sources.

## 3 Results

### 3.1. Comparative Experiments

To explore the performance differences among different models in semantic segmentation tasks for Thangka image central figures and ritual instruments, five mainstream segmentation models—STDC1 [4], STDC2 [4], DDRNET-S [3], PIDNET-S [5], and DDRNET [3]—were selected for comparative experiments.

The comparative experimental results are shown in Table 2. The proposed model in this chapter ranks first in both core metrics of mIoU (73.33%±0.12%) and mB-Fscore (40.06%±0.17%). Compared with the powerful baseline PIDNet-m, our model achieves a 3.13 percentage points higher mIoU and a 4.09 percentage points higher mB-Fscore. This clearly demonstrates that through our improvements, the model has achieved significant enhancements in segmentation accuracy and edge refinement. When compared with lightweight models such as DDRNet-s and the STDC series, our model exhibits even more prominent advantages.

In terms of parameter count, our model (29.418M) is comparable to PIDNet-m (28.761M) and considerably higher than DDRNet (20.299M). Although the parameter count is slightly higher than that of DDRNet-s and the STDC series, our model offers substantial accuracy advantages. In terms of inference speed, our model reaches 109.06 ± 0.52 FPS, leading the best speed performance among models with a similar parameter scale around 20M. It significantly outperforms DDRNet (80.31±0.38 FPS) and STDC2 (80.32±0.50 FPS), and also achieves a slight edge over PIDNet-s (107.35±0.51 FPS) and PIDNet-m (104.63±0.52 FPS). Instead of being extremely outstanding in a single indicator, the proposed model performs excellently across multiple metrics, achieving a balanced performance.

To evaluate the model performance in greater detail, Table 3 presents a comparison of IoU values across all categories.

From the perspective of regional segmentation accuracy (IoU), the improved model achieves significant improvements in the majority of ritual instrument categories such as "Vajra", "Scripture", and "Sword". This directly validates the correctness and necessity of the ECA-Pag and LGFM modules in enhancing the model's ability to extract features of multi-scale and complex-structured targets. Although there is a slight decrease in the IoU of large-scale targets like the "Main Deity", this is perfectly consistent with the expectations and trade-offs of this study: the core objective of this research is to address the segmentation challenges of detailed elements in Thangka images, and the optimization focus of the model is

**Table 2. Quantitative comparison results of related methods.**

| Modles | mIoU（%）↑ | mB-Fscore(%)↑ | Parameters(M） | FPS↑ |
|---|---|---|---|---|
| STDC1 [4] | 34.90±0.18 | 13.21±0.17 | 8.257 | 125.42±0.70 |
| STDC2 [4] | 31.61±0.16 | 11.59±0.10 | 12.306 | 80.32±0.50 |
| DDRNet-s[3] | 62.99±0.18 | 29.13±0.19 | 5.734 | 133.58±0.59 |
| PIDNet-s[5] | 64.40±0.14 | 30.28±0.10 | 7.865 | 107.35±0.51 |
| ourPIDNet-s | 64.49±0.14 | 31.28±0.10 | 7.865 | 92.38±0.38 |
| DDRNet [3] | 69.16±0.15 | 32.76±0.16 | 20.299 | 80.31±0.38 |
| PIDNet-m[5] | 70.20±0.17 | 35.97±0.13 | 28.761 | 104.63±0.52 |
| Ours | 73.33±0.12 | 40.06±0.17 | 29.418 | 109.06±0.52 |

**Table 3. Quantitative Comparison of IoU Metrics for Different Element Categories across Related Methods.**

| label | DDRNet [3] | PIDNet-m[5] | Ours |
|---|---|---|---|
| | IoU(%) | IoU(%) | IoU(%) |
| **Background** | 97.57 | 97.99 | 97.14 |
| **Main Deity** | 91.64 | 92.78 | 90.09 |
| **Vajra** | 37.18 | 47.79 | 58.91 |
| **Vase** | 86.71 | 87.74 | 87.23 |
| **Bowl** | 80.07 | 84.10 | 84.64 |
| **Dharma Wheel** | 66.76 | 74.61 | 73.14 |
| **Lotus** | 51.92 | 63.45 | 70.05 |
| **Scripture** | 58.06 | 70.01 | 81.82 |
| **Vajra Bell** | 54.54 | 58.23 | 59.04 |
| **Kapala Bowl** | 42.94 | 58.42 | 58.44 |
| **Sword** | 55.85 | 57.43 | 68.03 |
| **Umbrella** | 43.68 | 57.47 | 66.00 |
| **Khatvanga** | 51.09 | 61.76 | 69.13 |

inevitably tilted towards more challenging small-scale targets. The results demonstrate that our improvement strategy has accurately enhanced the model's performance on key difficult points, achieving the main goal of technical breakthrough.

As shown in Table 4, in terms of boundary segmentation quality (mB-Fscore), nearly all ritual instrument categories achieve prominent performance improvements—with the "Scripture" category achieving an impressive 23.33% increase—serve as the most compelling evidence of the success of this study. This strongly indicates that the collaborative design of the introduced LGFM module and the optimized detail branch (PConv3) is highly effective, successfully addressing the long-standing pain points of traditional models such as blurred and discontinuous edges of ritual instruments. As for the slight decrease in boundary scores for large-scale targets, this is not a flaw in the model design but an interpretable phenomenon resulting from the focused capability of the model: the model allocates stronger perceptual capacity to the edges of complex details, thereby making it more sensitive to minor variations in large-area smooth boundaries.

**Table 4. Quantitative comparison of B-Fscore metrics for different element categories across related methods.**

| label | DDRNet [3] | PIDNet-m[5] | Ours |
|---|---|---|---|
| | B-Fscore(%) | B-Fscore(%) | B-Fscore(%) |
| **Background** | 44.51 | 45.77 | 40.96 |
| **Main Deity** | 45.95 | 56.91 | 42.26 |
| **Vajra** | 25.30 | 27.09 | 37.89 |
| **Vase** | 63.25 | 59.81 | 60.08 |
| **Bowl** | 40.51 | 48.39 | 50.67 |
| **Dharma Wheel** | 32.66 | 36.87 | 35.96 |
| **Lotus** | 21.86 | 23.88 | 35.30 |
| **Scripture** | 17.88 | 21.19 | 44.52 |
| **Vajra Bell** | 26.38 | 38.70 | 40.52 |
| **Kapala Bowl** | 30.61 | 26.69 | 39.66 |
| **Sword** | 27.54 | 26.59 | 30.29 |
| **Umbrella** | 28.59 | 18.76 | 34.78 |
| **Khatvanga** | 20.62 | 35.18 | 27.18 |

As illustrated in Fig 6, an analysis of the semantic segmentation results on Thangka images reveals that the proposed PIDNet-based enhanced model significantly outperforms existing approaches—including STDC2, DDRNet, and the original PIDNet—in both segmentation accuracy and fine-detail preservation. The STDC2 model exhibits considerable false negatives and false positives when handling complex textures and cluttered backgrounds. Although DDRNet shows improvement in overall segmentation, it still lacks precision in fine-grained regions. PIDNet performs relatively well in capturing primary structures and some details, yet remains limited in modeling local intricate patterns and disentangling complex backgrounds.

In contrast, the enhanced model accurately captures fine details in Thangka images, producing sharp and well-defined segmentation boundaries and achieving superior overall performance. It effectively addresses key challenges inherent to Thangka imagery—such as intricate backgrounds, diverse textures, and highly detailed iconographic patterns—thereby offering robust technical support for the study and preservation of Thangka art. These visual observations from Fig 6 align with our highest reported mB-Fscore, providing intuitive and compelling evidence of the model's substantial advantage in boundary segmentation quality.

## 3.2. Ablation experiments

To explore the impact of the LGFM, PConv3, and ECA-Pag modules on model performance, ablation experiments were conducted using PIDNet-m as the baseline model. The training and testing environments were kept consistent with the comparative experiments, and all training parameters were identical to ensure the fairness of the experimental results.

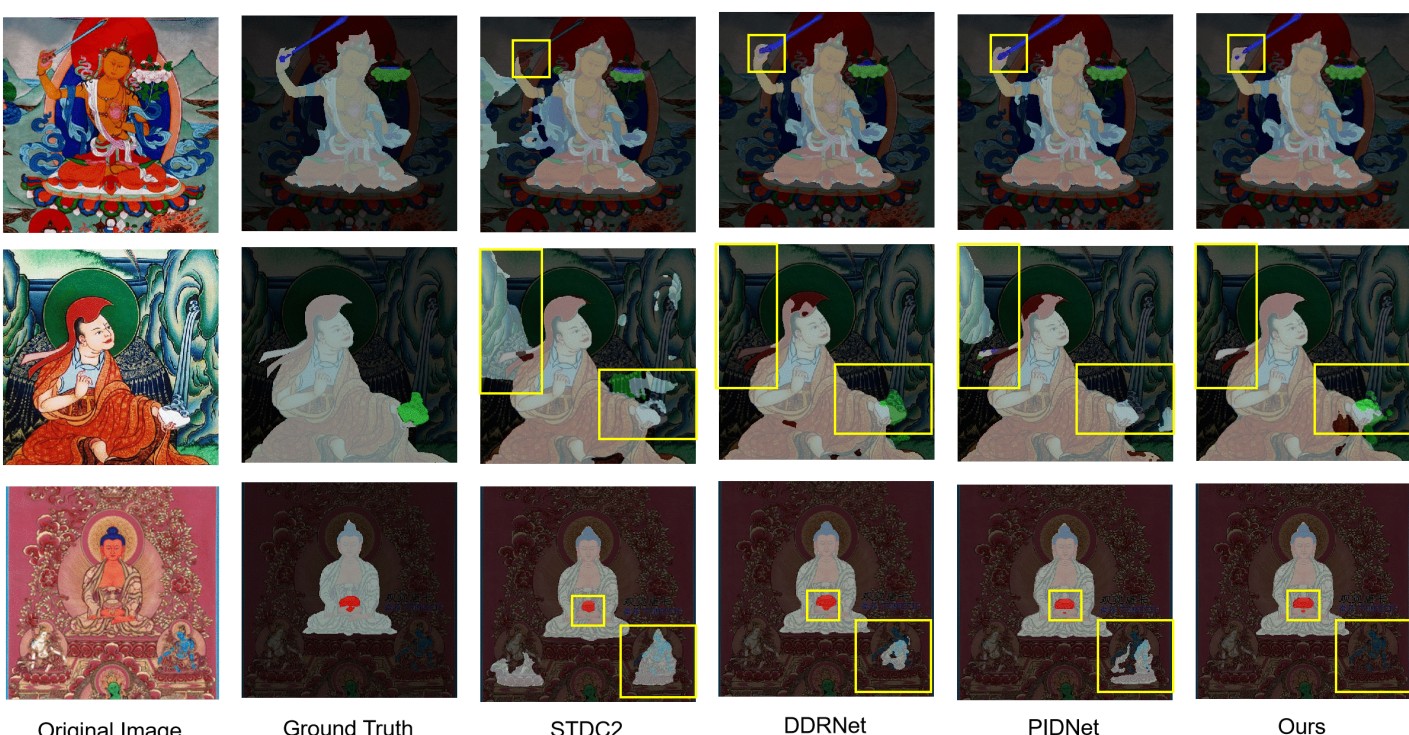

| Original Image | Ground Truth | STDC2 | DDRNet | PIDNet | Ours |

**Fig 6. Visualization results of related methods on the Thangka dataset.** [a]The rows show representative Thangka images with different compositions. From left to right: Original image, Ground truth, STDC2 segmentation result, DDRNet segmentation result, PIDNet segmentation result, Ours segmentation result. The proposed model exhibits clearer boundary segmentation and fewer false/missed detections for ritual instruments (e.g., lotus flower, vajra) and the main deity.

As shown in table 5, after introducing the ECA-Pag module alone (Model 2), the model's mIoU increased from 70.20%±0.17% (baseline, Model 1) to 70.76%±0.10%, with an improvement of 0.56 percentage points; the mB-Fscore was slightly adjusted from 35.97%±0.13% to 35.94%±0.10%, remaining basically stable. This indicates that ECA-Pag's channel recalibration effectively filters key semantic information for performance optimization, without increasing model parameters; the slight FPS drop is negligible due to the module's extremely low computational overhead.

By comparing Model 1 (baseline) and Model 3 (with PConv3), the introduction of PConv3 significantly increased the model's mIoU by 1.80 percentage points to 72.00%±0.11%, while the mB-Fscore increased by 0.49 percentage points to 36.46%±0.11%. More notably, the model's FPS soared from 104.63±0.52 to 117.12±0.23, a growth of approximately 11.94%. This fully verifies PConv3's dual advantages: its efficient local feature extraction boosts inference speed significantly, and the mask-constrained learning enables more efficient parameter utilization for discriminative boundary feature learning, realizing accuracy and speed synergy.

The LGFM module (Model 4) achieved the largest accuracy improvement among all independent components, with the mIoU increasing by 3.94 percentage points compared to the baseline to 74.14%±0.13%. This validates the crucial role of deep local-global feature fusion at the end of the P branch—the module can effectively model complex semantic correlations and scale differences in Thangka images, enhancing the robustness of target recognition. Meanwhile, the mB-Fscore stably increased to 36.41%±0.12%. The trade-off is a significant drop in FPS to 85.98±0.26 and an increase in parameter count to 29.224M, which is due to the certain computational complexity introduced by the COT self-attention and global context modeling in the LGFM module, consistent with the performance-efficiency trade-off law of complex feature fusion modules.

Integrating all three modules yields the final model (Model 5). Its mIoU is 73.33%±0.12%, 3.13 percentage points higher than the baseline; although slightly lower than Model 4 (with LGFM alone), it is much higher than other module combinations. More importantly, its mB-Fscore reaches 40.06%±0.17%, a substantial increase of 4.09 percentage points compared to the baseline, which is the highest among all models. This indicates the three modules' strong synergistic effect on boundary segmentation despite a slight regional segmentation trade-off: ECA-Pag's channel screening, PConv3's detail capture and LGFM's global modeling complement each other, significantly improving the segmentation accuracy of ritual instrument edges and other key fine details. Meanwhile, benefiting from the boost in inference speed by PConv3, the final model's FPS rebounds to 109.06±0.52, maintaining high-speed real-time inference capability; the parameter count is only 29.418M, with a slight increase compared to the baseline. In summary, the final model achieves the optimal comprehensive balance between segmentation accuracy (especially boundary quality), inference speed, and model lightweighting.

Table 6 reveals the nuanced impact of each proposed module on segmentation accuracy across diverse semantic categories. The LGFM module consistently delivers the most substantial gains for small, intricate ritual objects—most notably "Scripture" (+11.81% over baseline), "Sword" (+10.60%), and "Vajra" (+11.12%)—highlighting its strength in modeling long-range dependencies crucial for recognizing contextually sparse elements. In contrast, PConv3 shows a distinct advantage in preserving structural integrity for elongated or curved shapes: "Lotus" IoU jumps from 63.45% (baseline) to

**Table 5. Quantitative comparison results of the ablation experiments.**

| BaseLine | ECA-Pag | PConv3 | LGFM | mIoU (%)↑ | mB-Fscore (%)↑ | Parameters (M) | FPS↑ |
|---|---|---|---|---|---|---|---|
| 1 | – | – | – | 70.20±0.17 | 35.97±0.13 | 28.761 | 104.63±0.52 |
| 2 | √ | – | – | 70.76±0.10 | 35.94±0.10 | 28.761 | 101.50±0.24 |
| 3 | – | √ | – | 72.00±0.11 | 36.46±0.11 | 28.808 | 117.12±0.23 |
| 4 | – | – | √ | 74.14±0.13 | 36.41±0.12 | 29.224 | 85.98±0.26 |
| 5 | √ | √ | √ | 73.33±0.12 | 40.06±0.17 | 29.418 | 109.06±0.52 |

**Table 6. Quantitative Comparison of IoU Metrics for Different Element Categories in Ablation Experiments.**

| label | Baseline | Baseline+ECA-Pag | Baseline+PConv3 | Baseline+LGFM | Ours |
|---|---|---|---|---|---|
| | IoU(%) | IoU(%) | IoU(%) | IoU(%) | IoU(%) |
| **Background** | 97.99 | 98.21 | 97.74 | 97.14 | 98.08 |
| **Main Deity** | 92.78 | 93.51 | 91.98 | 90.09 | 92.99 |
| **Vajra** | 47.79 | 50.38 | 41.03 | 58.91 | 49.14 |
| **Vase** | 87.74 | 87.18 | 87.34 | 87.23 | 86.23 |
| **Bowl** | 84.10 | 83.51 | 84.41 | 84.64 | 82.93 |
| **Dharma Wheel** | 74.61 | 78.36 | 75.82 | 73.14 | 79.23 |
| **Lotus** | 63.45 | 62.56 | 70.34 | 70.05 | 65.48 |
| **Scripture** | 70.01 | 68.38 | 74.92 | 81.82 | 79.82 |
| **Vajra Bell** | 58.23 | 62.29 | 59.68 | 59.04 | 59.90 |
| **Kapala Bowl** | 58.42 | 57.86 | 59.50 | 58.44 | 61.96 |
| **Sword** | 57.43 | 55.56 | 65.37 | 68.03 | 64.84 |
| **Umbrella** | 57.47 | 54.30 | 60.52 | 66.00 | 57.36 |
| **Khatvanga** | 61.76 | 67.75 | 67.33 | 69.13 | 74.67 |

70.34%, and "Sword" improves by +7.94%, suggesting that its spatially constrained convolution better captures directional features. Interestingly, ECA-Pag yields modest global improvements but occasionally causes slight degradation on certain classes (e.g., "Umbrella" drops by 3.17%), indicating that channel-wise attention alone is insufficient without complementary structural modeling. The full model ("Ours") achieves the best overall balance: while it does not always match the peak performance of individual modules on specific classes (e.g., LGFM alone scores higher on "Scripture"), it maintains robust gains across nearly all categories—particularly excelling on "Khatvanga" (74.67%, + 12.91% over baseline)—demonstrating effective synergy among the three components.

The class-wise IoU results in Table 6 elucidate the distinct functional roles of each proposed module. The LGFM module yields the most pronounced improvements on small-scale, semantically sparse categories—specifically, "Scripture" (+11.81%), "Sword" (+10.60%), and "Vajra" (+11.12%) relative to the baseline—demonstrating its efficacy in capturing long-range contextual dependencies essential for accurate localization of intricate ritual objects. In contrast, PConv3 exhibits superior performance on structurally complex shapes such as "Lotus" (IoU: 70.34% vs. 63.45%) and "Sword" (65.37% vs. 57.43%), suggesting that its spatially selective convolution enhances the representation of directional and curvilinear features prevalent in Thangka iconography. The ECA-Pag module provides marginal global gains but occasionally induces minor performance degradation (e.g., −3.17% on "Umbrella"), indicating limited standalone impact on fine-grained segmentation. Notably, the full model ("Ours") achieves consistent improvements across the majority of classes, with a maximal gain of +12.91% on "Khatvanga", thereby validating the complementary nature of the three modules in jointly optimizing feature discrimination under the constraints of high visual complexity and category imbalance.

The boundary-aware metric B-Fscore (Table 7) provides critical insight into how each module refines object contours—a key challenge in Thangka segmentation. Notably, PConv3 dramatically enhances edge precision for texture-rich objects: "Scripture" B-Fscore surges from 21.19% (baseline) to 49.64%, and "Kapala Bowl" improves by +12.71%, confirming that its boundary-focused learning mechanism effectively sharpens fine strokes. LGFM also contributes significantly to boundary coherence, especially for geometrically complex items like "Vajra" (+3.71%) and "Dharma Wheel" (+2.09%), likely due to its ability to integrate global layout cues that guide local boundary placement. However, ECA-Pag exhibits mixed effects on boundaries—improving "Kapala Bowl" (+10.91%) but degrading "Main Deity" (−10.87%)—suggesting that indiscriminate channel reweighting may sometimes blur dominant foreground regions. Crucially, the complete

**Table 7. Quantitative Comparison of B-Fscore Metrics for Different Element Categories in Ablation Experiments.**

| label | Baseline | Baseline+ECA-Pag | Baseline+PConv3 | Baseline+LGFM | Ours |
|---|---|---|---|---|---|
| | B-Fscore (%) | B-Fscore(%) | B-Fscore(%) | B-Fscore(%) | B-Fscore (%) |
| Background | 45.77 | 44.70 | 40.95 | 42.27 | 40.96 |
| Main Deity | 56.91 | 46.04 | 42.01 | 42.91 | 42.26 |
| Vajra | 27.09 | 26.32 | 29.65 | 30.80 | 37.89 |
| Vase | 59.81 | 60.97 | 57.22 | 56.03 | 60.08 |
| Bowl | 48.39 | 50.33 | 45.12 | 44.78 | 50.67 |
| Dharma Wheel | 36.87 | 33.60 | 36.54 | 38.96 | 35.96 |
| Lotus | 23.88 | 26.49 | 22.40 | 26.06 | 35.30 |
| Scripture | 21.19 | 26.60 | 49.64 | 38.11 | 44.52 |
| Vajra Bell | 38.70 | 35.77 | 41.90 | 41.94 | 40.52 |
| Kapala Bowl | 26.69 | 37.60 | 39.40 | 38.67 | 39.66 |
| Sword | 26.59 | 29.35 | 29.36 | 26.48 | 30.29 |
| Umbrella | 18.76 | 26.22 | 24.34 | 23.13 | 34.78 |
| Khatvanga | 35.18 | 23.25 | 15.49 | 23.06 | 27.18 |

model ("Ours") achieves the highest B-Fscores on nearly all small/complex classes, with standout gains on "Scripture" (+23.33%), "Umbrella" (+16.02%), and "Vajra" (+10.80%). This confirms that the joint optimization of feature perception (ECA-Pag), boundary representation (PConv3), and contextual reasoning (LGFM) synergistically resolves the fragmented or over-smoothed edges that plague baseline models, directly addressing the core challenge of preserving Thangka's intricate linework and symbolic detail.

Fig 7 illustrates the changes in the evaluation metrics during the ablation experiments. By adding different modules to the baseline model, the model's performance is significantly improved. In particular, the model proposed in this paper not only performs excellently in terms of the mIoU but also achieves the best results in boundary prediction. This finding indicates that each proposed module has a positive effect on the model's overall performance, particularly in enhancing boundary prediction capabilities (Fig 8).

The visualization results of the ablation experiments are shown in Fig 9. By analyzing the visualization results in the figure, we can observe the gradual improvement in the semantic segmentation model's prediction effect on Thangka images with the independent introduction of different modules. Each column represents the independent introduction of a module, whereas the last column displays the final impact of comprehensively applying all the modules to the baseline model. The baseline PIDNet-m captures the general contours of main objects, but suffers from suboptimal detail extraction and background separation—e.g., blurred figure edges, unclear object-background boundaries and partial segmentation omissions.

Next, the ECA module is introduced (see the second column). The model's perception of key regions in the image, such as central figures and essential background elements, is significantly enhanced. In the second and third rows of the figure, background interference is significantly reduced, and the segmentation boundaries of key regions become more distinct, reducing missegmentation and omissions. Then, partial convolution is added to the baseline model (third column). This improvement significantly enhances the D branch's ability to capture detailed information. The figure shows that the fine textures and edges in the image are better preserved and processed. For example, in the second row, adding partial convolution to the baseline model improves the segmentation of figure clothing edges, making them more precise and accurate and thereby reducing blurring and omissions. The LGFM module is added to the P branch (fourth column). The fine details in the image, such as figure edges and clothing textures, are better preserved and presented. This indicates that the combination of local and global branches effectively captures more detailed information. In the second row of the

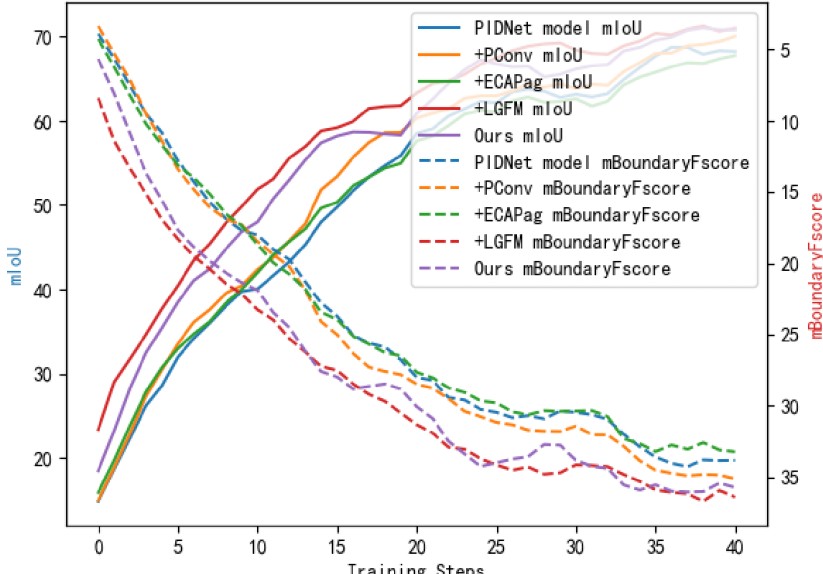

**Fig 7. Evaluation Metrics Variation Chart of Ablation Experiments.** [a]The chart shows the changes in mIoU and mB-Fscore with the addition of different modules. The combination of ECA-Pag, PConv3, and LGFM achieves the highest mB-Fscore, verifying the synergistic effect of the modules in boundary segmentation.

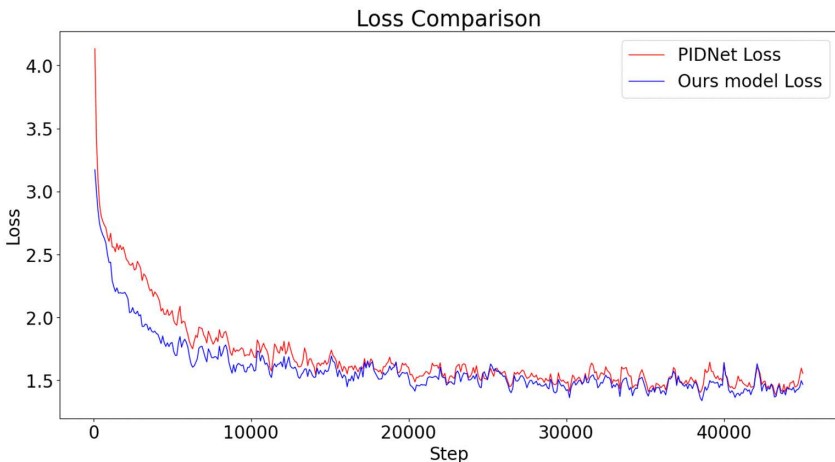

**Fig 8. Comparison Chart of Loss Functions between the Baseline Model and Our Model.** Figure 8 shows a comparison of the loss functions between the baseline model and the proposed model. Overall, the model proposed in this paper results in lower loss throughout the training process, indicating that it outperforms the PIDNet model at the same training steps, thereby verifying the effectiveness of the experiment.

figure, the segmentation details of the figure's face are more accurate. In the first row, the segmentation of the hand and ritual instrument contact part is also improved, making the segmentation result more complete and coherent.

Finally, all the modules are comprehensively applied. The final segmentation result is not only more accurate in key region identification but also more refined in detail processing, providing strong support for subsequent applications and further verifying the effectiveness and superiority of the proposed method.

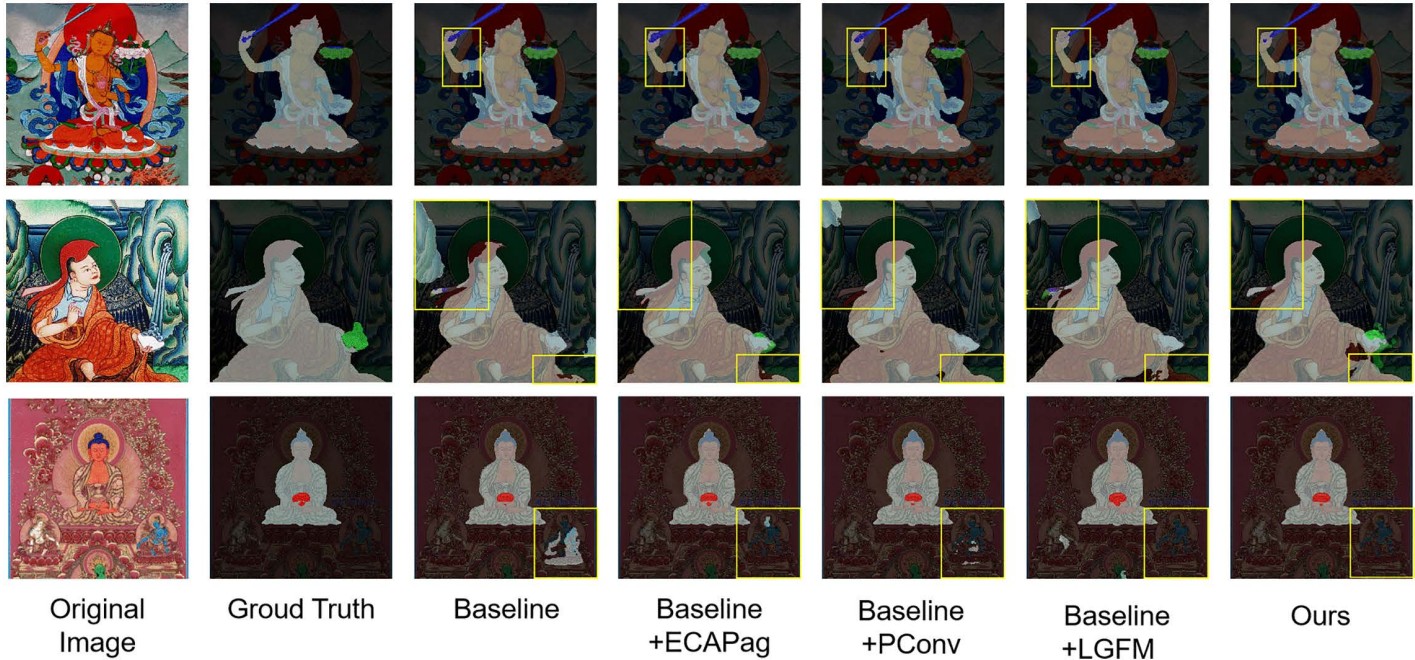

| Original Image | Groud Truth | Baseline | Baseline +ECAPag | Baseline +PConv | Baseline +LGFM | Ours |

**Fig 9. Visualization Results of Ablation Experiments.** [a]From left to right: Original image, Ground truth, Baseline (PIDNet-m) result, Baseline+ECA-Pag result, Baseline+PConv3 result, Baseline+LGFM result, Ours result. The proposed modules synergistically improve detail capture (e.g., clothing textures) and boundary clarity (e.g., ritual instrument edges).

### 3.3 Experimental comparison on cityscapes dataset

A comparative experiment was also conducted in this study using the public Cityscapes dataset [33], which is a semantic segmentation dataset dedicated to urban scenes. It contains a large number of high-resolution street-view images and covers various complex urban environments, including vehicles, pedestrians, roads, buildings, and other elements. The dataset includes annotations for 30 categories, among which 19 categories are used for the evaluation of semantic segmentation tasks. Owing to its rich annotations and high-resolution images, Cityscapes has been widely employed to evaluate the performance of semantic segmentation models, particularly in research related to autonomous driving and urban scene understanding.After the improvement of the PIDNet model, the proposed model also achieved outstanding performance on the Cityscapes dataset. The specific results are presented in Table 8:

To assess the generalization capability of the proposed architecture beyond the domain-specific Thangka dataset, we further evaluate its performance on the public Cityscapes benchmark, a standard semantic segmentation dataset for urban street scenes. Cityscapes comprises high-resolution images annotated across 19 semantic classes, covering complex elements such as vehicles, pedestrians, roads, and buildings, and is widely used in real-time vision tasks including autonomous driving.

As shown in Table 8, the improved model achieves an mIoU of 75.40%, outperforming not only lightweight real-time architectures—such as DDRNet (73.01%) and PIDNet (74.38%)—but also more computationally intensive transformer-based approaches. Notably, Swin Transformer-B, despite leveraging global self-attention and being trained with significantly longer schedules (160k iterations vs. 80k), attains only 52.4% mIoU under comparable input resolution (512×512), highlighting the efficiency gap between pure transformer models and convolutional designs optimized for real-time deployment. In contrast, our method maintains compatibility with the PIDNet backbone while introducing minimal computational overhead, thereby achieving state-of-the-art accuracy among real-time-capable models on Cityscapes.

**Table 8. Experimental Comparison on Cityscapes Dataset.**

| Models | Device | Crop Size | Lrschd Device | mlou(%)↑ |
|---|---|---|---|---|
| **Swin Transformer-B[34]** | V100 | 512*512 | 160000 | 52.4 |
| **STCNet [4]** | V100 | 512*1024 | 80000 | 71.82 |
| **DDRNet [3]** | RTX3090 | 1024*1024 | 80000 | 73.01 |
| **PIDNet [5]** | RTX3090 | 1024*1024 | 80000 | 74.38 |
| **Ours** | RTX3090 | 1024*1024 | 80000 | 75.40 |

† Results for Swin Transformer-B are reported at 512×512 input resolution due to high memory consumption of transformer-based models; all other methods are evaluated at 1024×1024 under identical training schedules (80k iterations).

The qualitative results in Fig 10 further corroborate these findings. Compared to DDRNet and PIDNet, our model produces sharper object boundaries (e.g., vehicle silhouettes), fewer false positives (e.g., background misclassified as pedestrians), and more coherent segmentation of small or distant objects (e.g., faraway pedestrians in column 2). These improvements stem from the synergistic effect of PConv3 (enhancing edge-sensitive features), ECA-Pag (refining channel-wise feature responses), and LGFM (integrating local detail with global layout)—mechanisms that prove effective not only in artistic imagery but also in naturalistic, cluttered urban environments.

Collectively, these results confirm that the proposed components generalize well across domains, offering robust performance gains without sacrificing real-time inference capability.

## Discussion

This study addresses the challenging real-time semantic segmentation of Thangka images, which feature intricate iconography, rich textures and complex symbolic compositions. To tackle this problem, we constructed a high-quality pixel-level annotated Thangka dataset and proposed an improved PIDNet architecture integrated with three key modules: the Local-Global Feature Fusion Module (LGFM), boundary-sensitive PConv3 block and ECA-Pag attention mechanism. Experimental results on the custom Thangka dataset show that the proposed model achieves state-of-the-art performance, with significant improvements in mIoU and boundary segmentation fidelity, while maintaining high real-time inference speed (109.06 FPS).

To verify the cross-domain generalization capability of the proposed approach, we further evaluated the model on the Cityscapes benchmark [33], a standard dataset for urban scene semantic segmentation. The model achieved an mIoU of 75.40%, outperforming existing real-time models such as DDRNet [3] and the original PIDNet [5]. This confirms that the three proposed modules encode transferable mechanisms for multi-scale context modeling and edge-aware feature learning, rather than being overfitted to Thangka's stylized features. Cityscapes was selected as the cross-domain testbed because it aligns with our core objective: balancing accuracy and efficiency under real-time constraints, and its focus on practical deployment scenarios (e.g., autonomous driving) makes it an ideal dataset to evaluate architectural generalization without compromising real-time performance.

Several limitations remain in this study. First, the current Thangka dataset, despite careful manual pixel-level annotation, is limited in scale and category diversity, especially for works with regional stylistic characteristics and rare ritual instruments. Second, the model's segmentation accuracy degrades in the presence of severe target occlusion or highly cluttered backgrounds, a common challenge for semantic segmentation in complex visual scenes. Third, the architecture is primarily optimized for symbolic, high-contrast artistic imagery, and may require adaptive adjustments for other cultural heritage modalities (e.g., faded ancient murals with blurred textures).

Future research will focus on three directions to address these limitations: expanding the Thangka dataset with diverse regional styles and rare elements to improve generalization; exploring occlusion-robust architectures via occlusion-aware

**Fig 10. Cityscapes dataset results visualization.** [a]**From top to bottom: Original image, Ground truth, DDRNet result, PIDNet result, Ours result.** The proposed model achieves clearer boundary segmentation (e.g., vehicle edges) and fewer false detections (e.g., background misclassified as pedestrians) in complex urban scenes.

attention and multi-scale feature enhancement modules for complex scenes; and extending the framework to other cultural heritage image segmentation tasks (e.g., murals, scroll paintings and stone carvings) to validate its adaptability and support digital preservation of more cultural heritage forms.

In summary, this study provides a robust and efficient solution for real-time Thangka image semantic segmentation, with the improved PIDNet achieving a favorable balance between segmentation accuracy, boundary fidelity and real-time inference. Notably, this work demonstrates that domain-specific innovations for cultural heritage image analysis can yield network architectures with meaningful cross-domain utility, enriching real-time semantic segmentation research in computer vision and providing a new technical approach for the digital preservation and research of Thangka cultural heritage.

## Supporting information

**S1 Fig. Visualization of attention characteristics in the LGFM module.**
(TIF)

## Author contributions

**Conceptualization:** Jiao Wu, Tiejun Wang.

**Formal analysis:** Jiao Wu, Tianjiao Duan.

**Methodology:** Jiao Wu, Xiaoyan Hu, LingMei Tao.

**Writing – original draft:** Yanjiao Wei.

**Writing – review & editing:** Tiejun Wang, Xiaoyan Hu, LingMei Tao.

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
