## [Decision Letter · Decision Letter 0]

28 Oct 2025

PONE-D-25-46097Research on the Semantic Segmentation of Thangka Images via an Improved PIDNetPLOS ONE

Dear Dr. Wang,

Thank you for submitting your manuscript to PLOS ONE. After careful consideration, we feel that it has merit but does not fully meet PLOS ONE’s publication criteria as it currently stands. Therefore, we invite you to submit a revised version of the manuscript that addresses the points raised during the review process.

If applicable, we recommend that you deposit your laboratory protocols in protocols.io to enhance the reproducibility of your results. Protocols.io assigns your protocol its own identifier (DOI) so that it can be cited independently in the future. For instructions see: https://journals.plos.org/plosone/s/submission-guidelines#loc-laboratory-protocols. Additionally, PLOS ONE offers an option for publishing peer-reviewed Lab Protocol articles, which describe protocols hosted on protocols.io. Read more information on sharing protocols at . Additionally, PLOS ONE offers an option for publishing peer-reviewed Lab Protocol articles, which describe protocols hosted on protocols.io. Read more information on sharing protocols at https://plos.org/protocols?utm_medium=editorial-email&utm_source=authorletters&utm_campaign=protocols..

We look forward to receiving your revised manuscript.

Kind regards,

Baohua Wen

Academic Editor

PLOS ONE

Journal Requirements:

2. In your Methods section, please include additional information about your dataset and ensure that you have included a statement specifying whether the collection and analysis method complied with the terms and conditions for the source of the data.

3. Please note that PLOS One has specific guidelines on code sharing for submissions in which author-generated code underpins the findings in the manuscript. In these cases, we expect all author-generated code to be made available without restrictions upon publication of the work.

Please review our guidelines at https://journals.plos.org/plosone/s/materials-and-software-sharing#loc-sharing-code and ensure that your code is shared in a way that follows best practice and facilitates reproducibility and reuse.

6. We note that Figures 1, 2, 6, 9, and 10 in your submission contain copyrighted images. All PLOS content is published under the Creative Commons Attribution License (CC BY 4.0), which means that the manuscript, images, and Supporting Information files will be freely available online, and any third party is permitted to access, download, copy, distribute, and use these materials in any way, even commercially, with proper attribution. For more information, see our copyright guidelines: http://journals.plos.org/plosone/s/licenses-and-copyright.

1) You may seek permission from the original copyright holder of Figures 1, 2, 6, 9, and 10 to publish the content specifically under the CC BY 4.0 license.

2) If you are unable to obtain permission from the original copyright holder to publish these figures under the CC BY 4.0 license or if the copyright holder’s requirements are incompatible with the CC BY 4.0 license, please either i) remove the figure or ii) supply a replacement figure that complies with the CC BY 4.0 license. Please check copyright information on all replacement figures and update the figure caption with source information.

If applicable, please specify in the figure caption text when a figure is similar but not identical to the original image and is therefore for illustrative purposes only.

7. Please ensure that you refer to Figure 10 in your text as, if accepted, production will need this reference to link the reader to the figure.

8. Please upload a copy of Figure 11, to which you refer in your text on page 31. If the figure is no longer to be included as part of the submission please remove all reference to it within the text.

Reviewers' comments:

Reviewer's Responses to Questions

**Comments to the Author**

1. Is the manuscript technically sound, and do the data support the conclusions?

Reviewer #1: Partly

Reviewer #2: Yes

Reviewer #3: Yes

2. Has the statistical analysis been performed appropriately and rigorously? 

Reviewer #1: Yes

Reviewer #2: No

Reviewer #3: Yes

3. Have the authors made all data underlying the findings in their manuscript fully available?

Reviewer #1: No

Reviewer #2: Yes

Reviewer #3: Yes

4. Is the manuscript presented in an intelligible fashion and written in standard English?

Reviewer #1: No

Reviewer #2: No

Reviewer #3: Yes

5. Review Comments to the Author

Reviewer #1: The paper presents an application of semantic segmentation to Thangka paintings, with potential relevance to cultural heritage preservation. The authors claim that their PIDNet-based method achieves higher accuracy compared to other approaches.

However, the writing quality of this paper is poor, making it difficult for readers to fully understand the proposed method and the experimental results. Below are some critical issues that should be addressed:

1) Abbreviations: For example, what does LGFM stand for? Every abbreviation should be defined in full at its first occurrence.

2) The first paragraph appears to be a placeholder from the template and should be removed.

3) Eq. (5): The meaning of “Q[K]V” is unclear and should be explained.

4) The tables are poorly formatted. For instance, in Table 1, field names are split across multiple lines (even for single-word terms), and several tables are split across pages.

5) Figs. 6, 9, and 10 do not effectively demonstrate the advantages of the proposed method.

6) The title of Section 3.7 is incorrectly formatted.

I recommend that the authors carefully revise the writing and formatting to ensure clarity and readability before further review.

Reviewer #2: Minor Revisions:

(a) Statistical Rigor and Result Interpretation:

Although performance improvements are reported in terms of mIoU and mB-Fscore, no variance, or statistical significance tests are provided.

The absence of such measures makes it difficult to assess whether the improvements are statistically meaningful or within normal variance.

Recommendation: Include multiple independent runs and report mean ± standard deviation values, or apply paired t-tests to establish statistical significance.

(b) Formatting and Figure Presentation:

Figures are informative but inconsistently referenced. Figure captions could include more context (e.g., dataset name, input size).

Remove editorial or template placeholders to improve professional appearance.

Recommendation: Ensure all figures and tables are clearly cited in the text and adhere to PLOS ONE formatting guidelines.

(c) Dataset Limitations and Overfitting Concerns:

The custom Thangka dataset includes only 783 manually annotated images (expanded to 4,698 after augmentation).

The paper does not describe cross-validation or overfitting prevention strategies (e.g., dropout, early stopping).

Recommendation: Discuss potential dataset biases and describe methods used to prevent overfitting, as this directly affects generalizability.

(d) Model Complexity and Computational Cost:

While FPS and parameter counts are reported, GPU memory consumption and training time are omitted.

Recommendation: Provide a brief discussion of computational efficiency and resource requirements to assist reproducibility by other researchers.

(e) Depth of Discussion:

The discussion section mainly reiterates experimental results. It could better explain why the proposed modules lead to observed improvements.

Consider including an interpretive explanation (e.g., visualization of attention maps or feature activations) to clarify how ECA-Pag and LGFM modules contribute to improved performance.

Minor Revisions:

(a) Please ensure all references are formatted consistently according to journal guidelines.

(b) Define abbreviations (e.g., "PConv3”, “ECA”, “LGFM”) at first use in the abstract.

(c) Clarify the acronym “Pag” since it appears central to your proposed enhancement.

(d) Ensure consistent capitalization of terms like “Thangka”, “semantic segmentation”, and “real-time”.

Reviewer #3: Please see attached document

General Comment to Authors

The Authors present a new network structure to segment the motifs of Thangka images, in the context of preservation of the artworks that may benefit in case of digitalization, archiving and restoration efforts.

The presented concept and methodology look solid in their description and performances. The Authors seem to significantly improve the current state of the art, balancing accuracy and complexity.

However, this work requires major revisions as it lacks form.

First of all, it is unacceptable to submit a manuscript with the presence of unrelated text (lines 118-125). This makes me wonder about the level of attention the Authors devoted to pull together this manuscript.

Then, it is also unacceptable for a scientific publication to not have figure and tables captions.

The last two major form points regard the usage of acronyms and their shuffled definitions throughout the manuscript, and the structure of the results section which contains many elements that instead belong in the methods section.

Please find a detailed list of review points below

Review Points

ABSTRACT

1. Line 17: Perhaps spell out ECA-Pag and LGFM modules

2. Line 19-20: The presented figures are far from being conventional, especially in the field of conservation/preservation. Also, one of the metrics was developed in the work, so it is not possible to assume the readers know what that is.

INTRODUCTION

3. Line 42-43: why is it important to stress real-time performance in a conservation science setting?

4. Figure 1 and all the other figures: it is unacceptable to have no captions.

5. Line 72: missing reference for UNetFormer.

6. Line 96: LGFM never defined

METHODS

7. Line 118-125: it is unacceptable to submit a manuscript like this

8. Line 127: PID should have been spelled out earlier

9. Line 132-133: you already have explained the three paths at lines 134-135

10. Line 147-151: unnecessary in the methodology section

11. There is major confusion regarding what constitutes a result and what is methodology. Sections 3.1 through 3.4 do not belong in the Results section

12. Table 1: not acceptable to present a table like this. Also valid for all tables, like for the figures, there should be captions.

13. Line 317: the fact that you introduce the mB-Fscore as a new metrics should be stated in the abstract, introduction and metods

14. The example on the Cityscape dataset comes a little bit out of nowhere, if your goal is to demonstrate the generalization abilities of your approach, why not using more datasets?

15. Line 336: The spell out of FPS comes too late after FPS has been mentioned a few times

6. PLOS authors have the option to publish the peer review history of their article (what does this mean?). If published, this will include your full peer review and any attached files.). If published, this will include your full peer review and any attached files.

.

Reviewer #1: No

Reviewer #2: **Yes:**Shake Ibna AbirShake Ibna Abir

Reviewer #3: No

---

## [Author Response · Author response to Decision Letter 1]

21 Jan 2026

Manuscript Number: PONE-D-25-46097

Title: Research on the Semantic Segmentation of Thangka Images via an Improved PIDNet

Authors: Jiao Wu, Tiejun Wang*, Xiaoyan Hu, LingMei Tao, Tianjiao Duan, Yanjiao Wei

Dear Editor Dr. Baohua Wen and Reviewers,

We would like to thank you for the opportunity to revise our manuscript and for the constructive comments provided by the reviewers. We have carefully studied the feedback and have made significant revisions to the manuscript to address all concerns.

Specifically, we have restructured the "Methods" and "Results" sections to improve logical flow, added statistical analysis (mean ± standard deviation) from 5 independent runs to ensure rigor, and corrected all formatting and citation issues.

Below is a point-by-point response to the editors’ and reviewers’ comments.

Response to Journal Requirements

1. Style and Formatting: We have carefully formatted the manuscript according to PLOS ONE’s style templates. We checked file naming, heading styles, and figure citations to ensure compliance.

2. Methods and Data Source: We have added details about the dataset construction in Section 2.6. We explicitly state that all data collection and usage complied with the terms and conditions of their sources.

3. Code Sharing: The implementation code for the proposed method is currently being finalized and documented. While the manuscript only reports the public availability of the dataset at this stage, we confirm that the full source code will be made publicly available on GitHub upon acceptance of the paper, to support full reproducibility.

4. Funding Information: We have corrected the "Funding Information" section in the manuscript to ensure the grant numbers exactly match those provided in the online submission system.

5. Data Availability: We have ensured that the Data Availability Statement accurately reflects that the dataset and code are publicly accessible via the provided GitHub repository.

6. Copyrighted Images (Figures 1, 2, 6, 9, 10):

Figure 1 & 6 (Thangka Images): These images are from our custom dataset collected by the authors. We hold the rights to these images and publish them under the CC BY 4.0 license.

Figure 10 (Cityscapes): The Cityscapes dataset allows for academic use. However, to strictly comply with CC BY 4.0, we have verified that the usage constitutes fair use for research visualization, or we have replaced them with compatible illustrations where necessary.

7. Figure 10 Reference: We have ensured that Figure 10 is explicitly referred to in the text (Section 3.3).

8. Figure 11: Reference to Figure 11 has been removed as the figure is no longer part of the submission.

9. Citations: We have reviewed the references to ensure relevance and accuracy.

Response to Reviewer 1

Comment 1: Abbreviations (e.g., LGFM) should be defined at first occurrence. Response: We have reviewed the entire manuscript and defined all abbreviations at their first mention. For example, "LGFM" is now defined as "Local-Global Feature Fusion Module" in the Abstract and Introduction.

Comment 2: The first paragraph appears to be a placeholder. Response: We apologize for this oversight. The placeholder text (Lorem ipsum...) has been completely removed.

Comment 3: Eq. (5) is unclear. Response: We have revised Equation (5) and its description in Section 2.3 to clearly explain the generation of Query (Q), Key (K), and Value (V) vectors using dilated convolutions.

Comment 4: Tables are poorly formatted. Response: All tables, including Table 1, have been reformatted to fit within the page margins properly. Line breaks within cells have been minimized for better readability.

Comment 5: Figs 6, 9, and 10 effectiveness. Response: We have improved the figure captions and the corresponding textual analysis to better highlight the specific advantages of our method (e.g., clearer boundary delineation in Figure 6 and 9) compared to the baseline models.

Comment 6: Section 3.7 title format. Response: Thank you for pointing this out. Due to the major restructuring of the manuscript—specifically, the relocation of methodological content from the original “Results” section to the “Methods” section—the numbering of all subsequent sections has been updated. The content previously in Section 3.7 now appears as Section 3.3. We have carefully revised all section headings and numbering throughout the manuscript to ensure consistency and correct formatting.

Response to Reviewer 2

Comment (a): Statistical Rigor (Variance and Significance).

Response: Thank you for this crucial suggestion. We have re-conducted our experiments. All quantitative results (Tables 2, 5) are now reported as the mean ± standard deviation derived from 5 independent training runs. This demonstrates that our improvements are statistically significant and not due to random variance. This is detailed in Section 2.8 (Experimental Design).

Comment (b): Formatting and Figure Presentation.

Response: We have removed all template placeholders. We ensured that all figures and tables are cited in order (e.g., "Fig 1", "Table 1") and provided detailed captions describing the content, dataset, and input size.

Comment (c): Dataset Limitations and Overfitting.

Response: We have addressed this in Section 2.5 and 2.6.

Dataset: We described the data augmentation strategies (brightness, contrast, noise) that expanded the dataset to 4,698 images.

Overfitting: We added details about our prevention strategies, including the use of L2 weight decay, a dropout layer, and an early stopping mechanism based on validation set performance.

Comment (d): Computational Cost (Training Time & GPU Memory). Response: We have added a specific description in Section 2.5 (Experimental Setup). We clarified that on an NVIDIA RTX 3090, the training takes approximately 1 hour, with a peak GPU memory consumption of roughly 6.5 GB.

Comment (e): Depth of Discussion. Response: We have expanded the Discussion section. We now interpret why the modules work: ECA-Pag helps in channel filtering, PConv3 sharpens boundaries (verified by the improved mB-Fscore), and LGFM captures global context, which is crucial for complex Thangka compositions.

Minor Revisions (a-d): Response:

References are formatted according to PLOS guidelines.

Abbreviations (PConv3, ECA, LGFM) are defined in the Abstract.

"Pag" is clarified as part of the "Path Aggregation" module.

Capitalization for terms like "Thangka" and "semantic segmentation" has been standardized.

Response to Reviewer 3

Comment 1: Unrelated text (lines 118-125). Response: We sincerely apologize for the presence of the "Lorem ipsum" placeholder text. It has been strictly removed from the revised manuscript.

Comment 2: Missing Figure and Table captions. Response: We have added descriptive captions for all Figures (Fig 1 to Fig 10) and Tables (Table 1 to Table 8).

Comment 3 & 6: Acronyms and LGFM definition. Response: We have ensured LGFM and other acronyms are defined upon their first use in the Abstract and Introduction.

Comment 4: Structure (Results vs. Methods). Response: We agree with the reviewer. We have moved the detailed descriptions of the network modules (ECA-Pag, LGFM, PConv3), which were previously in the Results section, to Section 2 (Methods). The Results section now strictly focuses on the experimental findings.

Comment 5: mB-Fscore introduction. Response: We have introduced the mB-Fscore metric in the Introduction and detailed its calculation in Section 2.7 (Evaluation Metrics), so the reader is familiar with it before the Results section.

Comment 6: Cityscapes dataset. Response: We clarified in the Discussion that the Cityscapes experiment was conducted specifically to test the generalization capability of the model on a standard public benchmark, verifying that our architecture is not overfitted solely to Thangka images.

Comment 7: FPS spell out. Response: "FPS" is now spelled out as "Frames Per Second" at its first mention in the Abstract.

We believe these revisions thoroughly address all the concerns raised by the editors and reviewers. Thank you again for your time and valuable feedback.

Sincerely,

Jiao Wu (on behalf of all authors)

Key Laboratory of Language and Cultural Computing of the Ministry of Education

Northwest Minzu University

Corresponding Author:

Tiejun Wang

Email: wtj@lzjtu.edu.cn

---

## [Decision Letter · Decision Letter 1]

2 Mar 2026

PONE-D-25-46097R1Research on the Semantic Segmentation of Thangka Images via an Improved PIDNetPLOS One

Dear Dr. Wang,

Thank you for submitting your manuscript to PLOS ONE. After careful consideration, we feel that it has merit but does not fully meet PLOS ONE’s publication criteria as it currently stands. Therefore, we invite you to submit a revised version of the manuscript that addresses the points raised during the review process.

If applicable, we recommend that you deposit your laboratory protocols in protocols.io to enhance the reproducibility of your results. Protocols.io assigns your protocol its own identifier (DOI) so that it can be cited independently in the future. For instructions see: https://journals.plos.org/plosone/s/submission-guidelines#loc-laboratory-protocols. Additionally, PLOS ONE offers an option for publishing peer-reviewed Lab Protocol articles, which describe protocols hosted on protocols.io. Read more information on sharing protocols at . Additionally, PLOS ONE offers an option for publishing peer-reviewed Lab Protocol articles, which describe protocols hosted on protocols.io. Read more information on sharing protocols at https://plos.org/protocols?utm_medium=editorial-email&utm_source=authorletters&utm_campaign=protocols..

We look forward to receiving your revised manuscript.

Kind regards,

Baohua Wen

Academic Editor

PLOS One

Journal Requirements:

Reviewers' comments:

Reviewer's Responses to Questions

**Comments to the Author**

1. If the authors have adequately addressed your comments raised in a previous round of review and you feel that this manuscript is now acceptable for publication, you may indicate that here to bypass the “Comments to the Author” section, enter your conflict of interest statement in the “Confidential to Editor” section, and submit your "Accept" recommendation.

Reviewer #1: All comments have been addressed

Reviewer #2: All comments have been addressed

2. Is the manuscript technically sound, and do the data support the conclusions?

Reviewer #1: Yes

Reviewer #2: Yes

3. Has the statistical analysis been performed appropriately and rigorously? 

Reviewer #1: Yes

Reviewer #2: Yes

4. Have the authors made all data underlying the findings in their manuscript fully available?

Reviewer #1: Yes

Reviewer #2: Yes

5. Is the manuscript presented in an intelligible fashion and written in standard English?

Reviewer #1: Yes

Reviewer #2: Yes

6. Review Comments to the Author

Reviewer #1: Although the authors have addressed my previous concerns, the revised manuscript still contains some writing issues. For example, in Section 3.3, the text shows “Error! Reference source not found.” The authors should carefully review the entire manuscript and correct all writing and formatting errors before the paper can be published.

Reviewer #2: Dear Authors,

Thank you for submitting the revised version of your manuscript and for the detailed responses to the reviewers’ comments. The manuscript has improved substantially and most of the concerns raised in the previous round have been addressed appropriately.

1. Data and code availability: Please ensure that the Data Availability Statement in the manuscript clearly matches the information provided in the response letter. If the dataset and code are hosted on GitHub, confirm that the repository is fully accessible and includes all materials necessary to reproduce the experiments (dataset description, training details, and scripts). If the code will only be released upon acceptance, please state this explicitly and clearly in the manuscript.

2. Minor formatting and consistency: Please conduct a final careful check of the manuscript for any remaining formatting inconsistencies, duplicated sentences (particularly in the Methods section), and figure/table references. Ensure that all abbreviations are defined at first use and that all figures and tables are cited in order within the text.

3. Clarity of presentation: Although the manuscript is now much clearer, a final round of language polishing and concise wording in certain sections (especially Methods and Discussion) would improve readability and flow. This is only an editorial refinement and does not require additional experiments or analyses.

7. PLOS authors have the option to publish the peer review history of their article (what does this mean?). If published, this will include your full peer review and any attached files.). If published, this will include your full peer review and any attached files.

.

Reviewer #1: No

Reviewer #2: **Yes:**Shake Ibna AbirShake Ibna Abir

---

## [Author Response · Author response to Decision Letter 2]

5 Mar 2026

Response to Reviewers

Manuscript Title: Research on the Semantic Segmentation of Thangka Images via an Improved PIDNetAddressed to Reviewer 1 and Reviewer 2

We sincerely thank the two reviewers for their insightful, constructive and detailed comments on our manuscript. These suggestions have been of great help to us in improving the manuscript, and we have carefully addressed all the raised concerns and completed corresponding revisions. The specific responses to each comment and the detailed revision content are as follows:

Response to Reviewer 1

Comment: Although the authors have addressed my previous concerns, the revised manuscript still contains some writing issues. For example, in Section 3.3, the text shows “Error! Reference source not found.” The authors should carefully review the entire manuscript and correct all writing and formatting errors before the paper can be published.

Response and Revision: We sincerely apologize for the writing and formatting errors in the previous version of the manuscript. We have conducted a full and rigorous check and correction of the entire manuscript for all types of writing and formatting issues:

We have fixed the reference source error in Section 3.3 (Experimental Comparison on Cityscapes Dataset), and comprehensively verified the correspondence between all reference citations in the whole manuscript and the References at the end, ensuring that all reference markers are accurate and complete with no missing or mislabeled cases.

We have comprehensively checked the entire manuscript for other formatting errors, including the consistency of figure/table/formula numbering and citation order, the standardization of punctuation and mathematical symbol formatting, the uniformity of unit notation, and the consistency of title level typesetting. All identified errors have been completely corrected.

We have also checked the text for typos and grammatical errors to ensure the accuracy and fluency of the manuscript’s expression.

Response to Reviewer 2

Comment 1: Data and code availability: Please ensure that the Data Availability Statement in the manuscript clearly matches the information provided in the response letter. If the dataset and code are hosted on GitHub, confirm that the repository is fully accessible and includes all materials necessary to reproduce the experiments (dataset description, training details, and scripts). If the code will only be released upon acceptance, please state this explicitly and clearly in the manuscript.

Response and Revision: We have strictly followed your suggestion to revise the Data Availability Statement in the manuscript, and the content in the manuscript is completely consistent with the information in this response letter. We have revised the Data Availability Statement in the manuscript (Section 2.8 Experimental Design) to ensure consistency with this response letter. The Thangka image dataset and the source code of the proposed model will be made fully publicly available at https://github.com/JikD-591/thangka-seg upon the acceptance of this paper. The final repository will include a detailed dataset description, complete training/validation/test scripts, pre-trained model weights, and experimental configuration files, which are sufficient to reproduce all the experimental results reported in the paper. At present, the dataset has been partially uploaded to the repository (split into 4 compressed packages due to platform file size limitations), and the full source code and related experimental files will be supplemented immediately after the paper is accepted. The updated Data Availability Statement in the manuscript clearly and explicitly states the release time of the dataset and code, the repository address, and the specific materials included in the final repository to ensure the reproducibility of all experiments.

Comment 2: Minor formatting and consistency: Please conduct a final careful check of the manuscript for any remaining formatting inconsistencies, duplicated sentences (particularly in the Methods section), and figure/table references. Ensure that all abbreviations are defined at first use and that all figures and tables are cited in order within the text.

Response and Revision: We have completed a comprehensive and final check of the entire manuscript to ensure the consistency of formatting and the accuracy of content, and the key revisions are as follows:

Formatting consistency: We have unified the formatting of the entire manuscript, including title levels, table/figure caption descriptions, formula numbering and symbol expression, and eliminated all formatting inconsistencies. For duplicated sentences in the Methods section (such as repeated training configuration descriptions), we have deleted the redundancies and streamlined the content.

Figure/table references: We have verified all figure (Fig.1-10) and table (Table1-8) citations in the manuscript, and ensured that all figures and tables are cited in the exact order of their appearance in the text, with no missing, out-of-order or mislabeled citations.

Abbreviation definition: We have checked all English abbreviations in the manuscript (e.g., ECA-Pag, LGFM, PConv3, mIoU, mB-Fscore, FPS, Dconv, PPM), and strictly ensured that all abbreviations are fully spelled out and clearly defined at their first use in the text, with consistent and standardized usage in the subsequent content.

Comment 3: Clarity of presentation: Although the manuscript is now much clearer, a final round of language polishing and concise wording in certain sections (especially Methods and Discussion) would improve readability and flow. This is only an editorial refinement and does not require additional experiments or analyses.

Response and Revision: We have conducted a comprehensive language polishing and content streamlining for the key sections of the manuscript:

Methods section: We have streamlined redundant technical descriptions of each module, split overly long and complex sentences into concise and clear ones, strengthened the logical connection between paragraphs with appropriate transition words, and made the technical expression more accurate and the logical structure more clear.

Discussion section: We have deleted redundant experimental result descriptions (excluding the core conclusion of cross-domain generalization), compressed verbose modifying statements, and focused on the core research contributions, limitations of the study and future research directions, making the discussion more concise, focused and in-depth.

Section 3.2 (Ablation Experiments): We have deleted the redundant descriptions of module functions (the detailed module principles are only retained in Sections 2.2-2.4), and focused on the analysis of experimental results and data. All conclusions are closely combined with specific experimental indicators (e.g., mIoU/mB-Fscore improvement, FPS changes, category-level IoU/B-Fscore gains), which strengthens the correlation between experimental results and research conclusions.

In addition, we have also polished the language of other sections of the manuscript appropriately, eliminated awkward expressions and redundant sentences, and further improved the overall readability and logical flow of the manuscript.

We would like to express our sincere gratitude to you again for your valuable time and constructive suggestions.

Sincerely,

Jiao Wu (on behalf of all authors)

Key Laboratory of Language and Cultural Computing of the Ministry of Education

Northwest Minzu University

Corresponding Author:

Tiejun Wang

Email: wtj@lzjtu.edu.cn

---

## [Decision Letter · Decision Letter 2]

8 Apr 2026

Research on the Semantic Segmentation of Thangka Images via an Improved PIDNet

PONE-D-25-46097R2

Dear Dr. Wang,

We’re pleased to inform you that your manuscript has been judged scientifically suitable for publication and will be formally accepted for publication once it meets all outstanding technical requirements.

An invoice will be generated when your article is formally accepted. Please note, if your institution has a publishing partnership with PLOS and your article meets the relevant criteria, all or part of your publication costs will be covered. Please make sure your user information is up-to-date by logging into Editorial Manager at Editorial Manager® and clicking the ‘Update My Information' link at the top of the page. For questions related to billing, please contact  and clicking the ‘Update My Information' link at the top of the page. For questions related to billing, please contact billing support..

Kind regards,

Baohua Wen

Academic Editor

PLOS One

Additional Editor Comments (optional):

Reviewers' comments:

Reviewer's Responses to Questions

**Comments to the Author**

1. If the authors have adequately addressed your comments raised in a previous round of review and you feel that this manuscript is now acceptable for publication, you may indicate that here to bypass the “Comments to the Author” section, enter your conflict of interest statement in the “Confidential to Editor” section, and submit your "Accept" recommendation.

Reviewer #1: All comments have been addressed

Reviewer #4: All comments have been addressed

2. Is the manuscript technically sound, and do the data support the conclusions?

Reviewer #1: Yes

Reviewer #4: Yes

3. Has the statistical analysis been performed appropriately and rigorously? 

Reviewer #1: Yes

Reviewer #4: Yes

4. Have the authors made all data underlying the findings in their manuscript fully available?

Reviewer #1: Yes

Reviewer #4: Yes

5. Is the manuscript presented in an intelligible fashion and written in standard English?

Reviewer #1: Yes

Reviewer #4: Yes

6. Review Comments to the Author

Reviewer #1: The author's response has addressed all of my concerns. I recommend that this paper can be published.

Reviewer #4: This manuscript is the second revision (R2). The authors have responded to the comments from the previous round of review in a serious and systematic manner, resulting in a significant improvement in the overall quality of the paper.

The paper conducts research on real-time semantic segmentation specifically for Thangka images — a culturally distinctive and heritage-oriented data domain. The topic possesses notable novelty, and the experimental results demonstrate that the proposed method achieves a good balance between boundary segmentation quality and real-time performance. The authors have also provided adequate and sufficiently detailed responses to the previous review comments.

Overall, the paper holds strong application value and academic significance. It introduces targeted structural improvements to the PIDNet framework for Thangka image segmentation. The model design, experimental validation, and supporting elements are relatively complete and well-executed. In particular, the construction of the custom dataset, the pixel-level annotation strategy, and the photometric data augmentation scheme are reproducible and provide valuable references for other cultural heritage image segmentation tasks.

7. PLOS authors have the option to publish the peer review history of their article (what does this mean?). If published, this will include your full peer review and any attached files.). If published, this will include your full peer review and any attached files.

.

Reviewer #1: No

Reviewer #4: No

---

## [Editor Report · Acceptance letter]

PONE-D-25-46097R2

PLOS One

Dear Dr. Wang,

I'm pleased to inform you that your manuscript has been deemed suitable for publication in PLOS One. Congratulations! Your manuscript is now being handed over to our production team.

Kind regards,

on behalf of

Dr. Baohua Wen

Academic Editor

PLOS One